# Toward Exploiting Second-Order Feature Statistics for Arbitrary Image Style Transfer

**DOI:** 10.3390/s22072611

**Published:** 2022-03-29

**Authors:** Hyun-Chul Choi

**Affiliations:** Intelligent Computer Vision Software Laboratory, Department of Electronic Engineering, Yeungnam University, 280 Daehak-Ro, Gyeongsan 38541, Gyeongbuk, Korea; pogary@ynu.ac.kr; Tel.: +82-53-810-2492

**Keywords:** image style transfer, second-order feature statistics, component-wise feature transform, mean and covariance loss, component-wise style control

## Abstract

Generating images of artistic style from input images, also known as image style transfer, has been improved in the quality of output style and the speed of image generation since deep neural networks have been applied in the field of computer vision research. However, the previous approaches used feature alignment techniques that were too simple in their transform layer to cover the characteristics of style features of images. In addition, they used an inconsistent combination of transform layers and loss functions in the training phase to embed arbitrary styles in a decoder network. To overcome these shortcomings, the second-order statistics of the encoded features are exploited to build an optimal arbitrary image style transfer technique. First, a new correlation-aware loss and a correlation-aware feature alignment technique are proposed. Using this consistent combination of loss and feature alignment methods strongly matches the second-order statistics of content features to those of the target-style features and, accordingly, the style capacity of the decoder network is increased. Secondly, a new component-wise style controlling method is proposed. This method can generate various styles from one or several style images by using style-specific components from second-order feature statistics. We experimentally prove that the proposed method achieves improvements in both the style capacity of the decoder network and the style variety without losing the ability of real-time processing (less than 200 ms) on Graphics Processing Unit (GPU) devices.

## 1. Introduction

Generating an image of artistic style from an input content image and a target-style image, also known as image style transfer, is one of the popular research topics in computer vision. Classical image style transfer [1] was performed by transforming the responses of human-designed filters from a content image to those on a style image. As the most of computer-vision-related research topics, image style transfer also recently became an application of deep neural networks (DNN) and has been improved in image quality [2,3,4,5] and processing speed [6,7,8].

Recently, convolutional neural networks (CNN) [9,10,11] achieved q capacity of multiple style in a network. They aligned the second-order statistics of the encoded feature map of a content image into that of a target style image in their modified instance normalization layer. In the background of those methods, it was assumed that the style of an image can be represented as the simplified second-order statistics, i.e., mean and standard deviation, of its encoded feature through Visual Geometry Group (VGG) encoder [12].

However, their feature alignment techniques did not consider the existing correlation between channels of the encoded feature map, where the correlation represents an important factor of a style, i.e., the co-occurrence of different patterns on an image. Some methods [9,10] used correlation-aware loss in decoder network training but their losses are not consistent with their feature alignment methods. Another method [13] did a correlation-aware feature alignment but still used inconsistent loss in decoder learning.

Since this inconsistency may degrade the style capacity of the decoder network or the quality of output style in the task of image style transfer, the correlation among channels of the feature map should be considered consistently in both feature alignment and loss calculation. Kalischek et al. [14] used a new loss to match higher-order moments but their method is optimization-based style transfer and has no consistent style transformer. Kim and Choi [15] removed correlations between feature channels for faster style transfer but they did not deal with style component control.

In this paper, the second-order statistics of the encoded feature maps are exploited to build an optimal style transfer with a neural network of encoder/transformer/decoder architecture as shown in Figure 1. First, a new style loss (style loss of the red box in Figure 1), i.e., (mean + covariance) loss, is used to improve style distinctiveness in the generated images. The proposed style loss considers both average impression (mean loss) and co-occurrence (covariance loss) of style patterns on an image. In addition, The proposed style loss is consistent with the very recent correlation-aware feature alignment technique [13,16].

Therefore, these consistent feature alignment and style loss are used to train a decoder network for higher style capacity and lower style loss. Secondly, style components (component-wise stylization of the red box in Figure 1), i.e., average and variations, from the second-order feature statistics are utilized to component-wise control the style of an output image. This enables a style transfer network to generate images of various styles from a single target-style image or component-wise style combination of several target-style images.

By doing a bundle of experiments, it is verified that using both correlation-aware feature transformer and loss achieves an improvement in style capacity of a network and competitive qualities in the generated styles and that the proposed component-wise feature transform achieves various styles in the generated output image from the given input content and target style images without losing the realtime speed, the ability of multi-style interpolating, and style strength controlling of the previous methods [9,10,11,13].

## 2. Background

The first neural-network-based image stylization was proposed in [2]. This used a pre-trained feature extractor of VGG-net [12] as the filter banks for extracting content and style features of an image. The responses of deeper convolution layers and lower convolution layers of VGG-net were assumed to represent content and style features of an image respectively. Based on this assumption, the L2 distance between the content features of input content image and output stylized image is defined as content loss, and the Frobenius distance between Gram matrices, also known as the Gram loss, of the features as a style loss. Then, a gradient-based optimization method was used to find pixel values of the output stylized image, which minimizes both the content and style losses.

This method achieved a plausible quality of output stylized images with any pair of content and target style images through a time-consuming optimization of output pixel values. Later, photo-realistic style transfer with conceptual segmentation information [4], histogram loss [5], the maximum mean discrepancy (MMD) [17] and perceptual factor control [3] were proposed to improve the quality of output stylized images.

Two very similar methods [6,7] achieved fast image style transfer by moving the previous online task of pixel value optimization into an offline task of network learning by inserting a CNN-based encoder/decoder network between the input content image and output stylized image. The network was trained to minimize the sum of content and style losses for a specific style embedding. After training a network, output images of the embedded style were generated in real-time by simply feeding content images to the trained network. Ulyanov et al. [8] further improved the quality of the output stylized images by changing the batch normalization (BN) layer [18] into an instance normalization (IN) layer in their encoder–decoder network.

Dumoulin et al. [9] proposed a method to embed multiple styles in a network. They introduced a modified IN layer, i.e., a conditional instance normalization (CIN) layer, which transformed the normalized features from an IN layer by using style-specific offset and scale parameters. Their method achieved a learning capability of several tens of styles and enabled selecting a specific style among the trained styles in a network. An updated method [10] adopted an additional inception network to generate the style-specific parameters for CIN layers from an input target-style image and achieved style transfer for arbitrary target styles.

Huang and Belongie [11] introduced an adaptive instance normalization (AdaIN) layer that used the mean and standard deviation calculated from the target style features as the linear transform parameters. AdaIN, incorporating the (mean + standard deviation) loss, resulted in arbitrary style transfer. They also simplified the learning complexity of the network by adopting VGG-net as its fixed encoder.

Recent domain adaptation techniques [16,19] showed that correlation alignment (CORAL) [16] in second-order feature statistics improved the performance of a trained network in image classification for unseen domains and that the performance was further improved by training a feature extractor with a correlation loss (Deep CORAL) [16].

This method can be also applied to image style transfer, and Li et al. [13] used CORAL with a different name of whitening and coloring transform (WCT) in the transformer layer of a style transfer network. They additionally adopted cascade networks and achieved correlation-aware and multi-scale style transfer. However, they used image reconstruction loss for training decoder networks without considering consistency with their transformer layer.

Some approaches based on generative adversarial networks (GANs) [20,21] also dealt with image style transfer as an application of their image-to-image translation task. Pix2pix [20] used conditional GAN (cGAN) to learn a generator and a discriminator simultaneously from a set of source and target image pairs. CycleGAN [21] relieved the requirement of the source and target image pairs in the cGAN training phase by utilizing cycle consistency between the source image and cyclic reconstructed image. However, there was no consideration of multiple style embedding and style control in their methods.

## 3. Method

In this section, each part of exploiting second-order feature statistics, i.e., the (mean + covariance) loss and component-wise style control, is described.

### 3.1. (Mean + Covariance) Loss: Losses for Style Transfer Revisited

Here, we show that the original loss of neural style [2] lacks style distinctiveness. Then, a new style loss, i.e., (mean + covariance) loss, is defined to improve the style distinctiveness of the generated image and to increase the style embedding capacity of a network.

Given two tensors *X*, *Y*
$\in R^{C\times (H\times W)}$ that have *C*-channel responses in a convolution layer of H×W pixel size, Gram matrix $G\in R^{C\times C}$ is defined as a correlation matrix of channels in the tensor, and Gram loss $L_{gram}$, the well-known style loss, is defined as the Frobenius distance (${\left||.|\right|}_{F}$) of two Gram matrices as Equation (Equation 1) [2,3,4,6,7,8,9,10].
(1)$$L_{gram}=||G\left(X\right)-{G\left(Y\right)||}_{F},\;\;G\left(X\right)=E\left[XX^{T}\right]\in R^{C\times C},$$
(2)$$\begin{array}{cc} V(X) & =E[\left(X-u_{X}\right){\left(X-u_{X}\right)}^{T}]\\ \; & =E\left[XX^{T}\right]-u_{X}{u_{X}}^{T}\in R^{C\times C},\;\;u_{X}=E\left[X\right]\in R^{C}. \end{array}$$
A Gram matrix can also be expressed alternatively as the sum of covariance V(X) and mean correlation $u_{X}{u_{X}}^{T}$ as Equation (Equation 3) derived from the second-order statistics of the tensor (Equation (Equation 2)). Here, the mean response $u_{X}$ can be considered as a distinctive factor of style that represents how the style of a drawing, such as te strokes and patterns, is on average across the whole area. The covariance V(X) can be another factor of style as the variety of patterns and strokes from the average.
(3)$$G\left(X\right)=E\left[XX^{T}\right]=V\left(X\right)+u_{X}{u_{X}}^{T}.$$
According to Equation (Equation 3), the Gram matrix cannot differentiate two different tensors that have different covariances (V(X)≠V(Y)) and mean responses ($u_{X}\neq u_{Y}$) but occasionally the same Gram matrices (G(X)=G(Y)). Therefore, we concluded that Gram loss itself is not an appropriate measurement of style similarity. Instead, a new style loss $L_{style}$ between two different tensors *X* and *Y* as the weighted summation of mean loss $L_{mean}$ and covariance loss $L_{cov}$ with a scalar weight $w_{mc}$ are defined as shown in Equation (Equation 4). The new style loss considers similarities in both the average style and style variation. $w_{mc}=1.0$ is used in the experiments but can be adjustable.
(4)$$\begin{array}{cc} L_{style} & =L_{mean}+w_{mc}L_{cov},\\ L_{mean} & ={\left|\right|u_{X}-u_{Y}{\left|\right|}_{2}}^{2},\\ L_{cov} & =\sqrt{||V\left(X\right)-{V\left(Y\right)||}_{F}}. \end{array}$$

The proposed style loss can be understood as a generalized version of the (mean + standard deviation) loss of [11], which has no consideration of the correlation between channels of the feature map. The proposed style loss considers the channel correlation by using the covariance loss $L_{cov}$ instead of the standard deviation loss [11]. Here, the square root of the Frobenius distance of covariances is used to match its dimension to the standard deviation. As the red boxes in Figure 1, the proposed style loss (Equation (Equation 4)) is incorporated into the proposed feature transform method described in the following section for training the decoder network, where learning a network with a consistent loss and feature alignment method is proven to improve the domain adaptation performance in the image classification task [16,19].

For the content loss $L_{content}$, the previously used L2 loss [2,9,13] between the feature maps of content and output images is used as shown in Figure 1 (loss in black box). This content loss helps the decoder maintain the perceptual content of the content image in generating a stylized image.

Finally, the total loss for decoder training is represented as a weighted summation of those two losses (Equation (Equation 5)). $w_{s}=50$ is used in the experiments.
(5)$$L_{total}=L_{content}+w_{s}\cdot L_{style}$$

### 3.2. Component-Wise Feature Transform (CWFT)

If the task of style transfer from an arbitrary content image to an arbitrary target style is assumed as a problem of how to adjust a given network to an unseen domain of the given images, an appropriate domain adaptation technique is required. CORAL [16] (or WCT [13]) is a simple but effective feature alignment technique for this purpose. Given two feature maps $X,\;Y\in R^{C\times (H\times W)}$ that have *C*-channels and H×W pixels, CORAL transforms the second-order statistics of *X* into the zero-mean and unit-variance of X|0 through Equation (Equation 6) and then into that of X|Y through Equation (Equation 7) by using covariances ($V\left(X\right),\;V\left(Y\right)\in R^{C\times C}$) and means ($\mu_{X},\;\mu_{Y}\in R^{C}$) in Equation (Equation 8).
(6)$$\begin{array}{cc} X|0 & =U_{X}{S_{X}}^{-1}{U_{X}}^{T}\left(X-\mu_{X}\right),\\ S_{X} & =diag({\lambda_{X}}_{1},\;\cdots ,\;{\lambda_{X}}_{C}), \end{array}$$
(7)$$\begin{array}{cc} X|Y & =U_{Y}S_{Y}{U_{Y}}^{T}X|0+\mu_{Y},\\ S_{Y} & =diag({\lambda_{Y}}_{1},\;\cdots ,\;{\lambda_{Y}}_{C}), \end{array}$$
(8)$$\begin{array}{cc} V\left(X\right)=U_{X}{S_{X}}^{2}{U_{X}}^{T}, & \;\;V\left(Y\right)=U_{Y}{S_{Y}}^{2}{U_{Y}}^{T},\\ u_{X}=E\left[X\right], & \;\;u_{Y}=E\left[Y\right], \end{array}$$
where $\lambda_{X}$ and $\lambda_{Y}$ represent the square root of eigenvalues of V(X) and V(Y) respectively.

The proposed component-wise feature transform (CWFT) utilizes each column of the unitary matrix $U_{Y}$ in Equation (Equation 8) as the independent style components of *Y* to generate various styles from a target style image. As shown in Figure 1, after the correlation-aware style normalization step of Equation (Equation 6), the normalized feature X|0 is stylized into the target style of the encoded feature *Y* of style image $I_{s}$ in a similar manner of CORAL stylization (Equation (Equation 7)) but component-wise as Equation (Equation 9).
(9)$$X|Y\left(\beta_{0..C}\right)=U_{Y}{\hat{S}}_{Y}\left(\beta_{1..C}\right){U_{Y}}^{T}X|0+\beta_{0}\mu_{Y},$$
(10)$$\begin{array}{c} {\hat{S}}_{Y}\left(\beta_{1..C}\right)=diag\left(\beta_{1}{\lambda_{Y}}_{1},\;\cdots ,\;\beta_{C}{\lambda_{Y}}_{C}\right),\\ 0.0\leq \beta_{j}\leq 1.0\;for\;j=0..C, \end{array}$$
where $\beta_{j}$s are weights for independent style components (columns of $U_{Y}$) and average style $\mu_{Y}$. The strengths of the style components are independently controlled by these style component weights $\beta_{j}$s. Here, an output image is fully stylized when $\beta_{j}=1.0$ for all *j*s and partially stylized when $\beta_{j}<1.0$ for any *j*.

Unlike the previous fast style transfer methods [9,10,11,13], which can only control whole style, the proposed method can control each component of style independently between a content image and a target style image (Equation (Equation 11)) or between *N* target style images (Equation (Equation 12)) by using linear interpolation with the style strength parameters α or $\alpha_{i}$ incorporating component-wise stylization (Equation (Equation 9)) of CWFT.
(11)X|Y(α)=αX|Y+(1−α)X,0.0≤α≤1.0,
(12)$$\begin{array}{c} X|Y_{1\ldots N}\left(\alpha_{1\ldots N}\right)=\Sigma_{i=1}^{N}\alpha_{i}X|Y_{i}+\left(1-\Sigma_{i=1}^{N}\alpha_{i}\right)X,\\ 0.0\leq \alpha_{i}\leq 1.0\;for\;i=1\ldots N,\;\Sigma_{i=1}^{N}\alpha_{i}<1.0. \end{array}$$

## 4. Experiments

### 4.1. Experimental Setup

The same architecture of the network of [11] with a fixed pre-trained VGG16 encoder [12] and a trainable mirrored VGG16 decoder were used for all experiments because the simple structure is effective to see the effect of considering independent style components in the proposed component-wise feature transformer (CWFT) and the new style loss. Two decoder networks with CWFT were trained, one with a small style dataset that consists of 22 drawings and the other with a large style dataset, painter by numbers [22] of 79,433 drawings.

The MS-COCO dataset [23] of 82,783 pictures was used as the content dataset for both two networks. The network trained with the small style dataset is for efficiently verifying the effect of the new style loss and CWFT, the other network trained with a large style dataset is for verifying generalization performance of the proposed method as the number of embedded styles increases. All images were resized to 256 pixels on the shorter side for both training and testing and randomly cropped into 240×240 pixels only for training to prevent boundary artifacts without losing the style statistics of the image.

As in [6], the set of responses of (ReLU1_2,ReLU2_2,ReLU3_3,ReLU4_3) layers was used as the style feature for calculating the style loss and the response of $relu3_{3}$ layer as the content feature for calculating content loss and transforming. The training iteration went until four epochs with batch size 4 of the random pair of content and style images, with Adam optimizer [24] and learning rate of $10^{-4}$. Pytorch framework with CUDA and CuDNN was used on NVIDIA 1080 TI GPU for experiments.

### 4.2. Performace of the (Mean + Covariance) Style Loss

To compare the performance of the new style loss with the previously used losses, several networks were trained with the previous feature alignment methods, i.e., CIN [9], AdaIN [11] and CORAL [19] (or WCT [13]), by using a style loss among Gram loss [2,9], (mean + standard deviation) loss [11], reconstruction loss [13] and the proposed (mean + covariance) loss.

Figure 2 shows some output stylized images using networks of a feature alignment method and a style loss trained with a small style dataset of Section 4.1. As the first and second rows of Figure 2 show, both CIN and AdaIN generated style-transferred images of high style quality but with artifacts of unnatural patterns (red dashed circles) on the sky region in the images for the first target style. Those artifacts do not exist in the images of the (mean + covariance) style loss on the third and fourth rows of Figure 2. By considering the average response and inter-channel correlation independently in style loss, the proposed style loss helps to generate only reasonably correlated style patterns, and this resulted in diminishing the artifacts that occurred in the CIN or AdaIN methods with their original style losses, such as Gram loss or (mean + standard deviation) loss.

Several networks were trained with a large style dataset of Section 4.1 for arbitrary style transfer to analyze the generalization performance of the proposed style loss. Figure 3 shows some output stylized images from the networks. The images on the first and second rows are from the networks with AdaIN + (mean + standard deviation) loss [11] and AdaIN + (mean + covariance) loss. As AdaIN matches the mean and standard deviation of feature maps from content image to style image, changing style loss from (mean + standard deviation) to (mean + covariance) appears to not affect the output images.

However, with WCT [13], which matches the mean and covariance of feature maps, using the (mean + covariance) loss resulted in a texture and color tone of the output images that was more similar to the style images (the last row of Figure 3) than with AdaIN. This improvement is more clear when comparing to WCT + reconstruction loss [13] (the third row of Figure 3). The image on the third row of Figure 3 shows black colored blobs while the images with the (mean + covariance) loss on the last row of Figure 3 do not have this kind of artifact.

For quantitative analysis, the losses of the networks were measured as the number of training data increased. These are presented in Figure 4. As the number of training data varied from $10^{1}$ to $10^{4}$, total loss (Equation (Equation 5)) also increased (the total loss in Figure 4) because the network has to embed more styles in itself. Here, using the (mean + covariance) loss with WCT, which is consistent with the proposed style loss, showed the lowest total loss (red line) compared with the other methods with AdaIN (blue and green lines). This means that using the consistent pair of correlation-aware loss and feature transform layers makes the network embed a greater number of styles at the same loss (performance).

This loss reduction is mainly for the style loss ((mean + covariance) loss in Figure 4) and, more specifically, the covariance loss (covariance loss in Figure 4) rather than the mean loss (mean loss in Figure 4). This indicates that either consistent or inconsistent usage of the feature transform layer and loss has a similar stylization quality on average (the first, the second and the last rows of Figure 4) but that using consistent pairs of correlation-aware feature transform layers (WCT) and loss (mean + covariance) transfers subtle changes from the average style better as described in the previous paragraph.

Of course, using AdaIN with (mean + standard deviation) loss achieved the best standard deviation loss (blue line on standard deviation loss in Figure 4) because it directly matched the mean and standard deviation in AdaIN and the optimized its decoder network in the aspects of mean and standard deviation loss. However, based on the covariance loss, this combination of correlation-unaware transform layer and loss could not transfer the correlations between feature map channels unlike WCT with the (mean + covariance) loss.

### 4.3. Results of Component-Wise Style Transfer and Control

To verify the style transferring ability of the proposed component-wise feature transformer (CWFT) described in Section 3.2, the output images through the trained decoder network corresponding to the content image features of the proposed stylizing procedure with several target-style images are presented in Figure 5. The first row of Figure 5 shows a content image and its style normalized features through the decoder network trained with a small style dataset.

The style normalized image still has some rough noise patterns with a weak content silhouette. The remaining rows of Figure 5 show examples of the stylized images of average style and their variations with some style components for several style images in the training dataset. Although the images of average style themselves have a specific style, the style components of lower eigenvalues (+0–20% style components) give subtle variations, and the style components of higher eigenvalues (+80–100% style components) give large variations to the stylized images.

If all $\beta_{j}$s = 1 in Equations (Equation 7) and (Equation 12) are used, then style strength control can be achieved as in the previous methods [11,13]. Figure 6 shows some examples of style strength control. In addition, the proposed method can generate stylized images with different strengths for style components as shown in Figure 7. $\beta_{1}$ is the strength of style components corresponding to upper 50% of eigenvalues and $\beta_{2}$ is that for the lower 50% of eigenvalues. The images shows that the stylized image varies from the average stylized image (top-left on Figure 7) to the fully stylized image (bottom-right on Figure 7) as the two parameters change.

As $\beta_{1}$ changes from 0.0 to 1.0, the stylized image has stronger colors and blob shapes of buildings (the first column of Figure 7). As $\beta_{2}$ changes from 0.0 to 1.0, the stylized image has stronger patters in sky and clear shapes of buildings (the first row of Figure 7). The images on the diagonal ($\beta_{1}=\beta_{2}$) can be obtained with the previous style strength control while a bunch of images of different styles on the whole rectangle for any combinations of $\beta_{1}$, and $\beta_{2}$ can be obtained with the proposed component-wise style control.

Figure 8 shows an example of component-wise style interpolation between two styles. Here, Equations (Equation 9) and (Equation 12) were used to interpolate the styles of top-right image and bottom-left image. While the previous style interpolation technique can makes only the image on diagonal of Figure 8 because it uses the same strength parameter for all style components (β=1.0), the proposed component-wise style control uses two different parameters ($\beta_{1lo}$ and $\beta_{1up}$ for the top-right style, $\beta_{2lo}$ and $\beta_{2up}$ for the bottom-left style, $\beta_{1lo}+\beta_{2lo}=1.0$, $\beta_{1up}+\beta_{2up}=1.0$, α=1.0) for the style components and resulted in various combinations of two styles of Figure 8. The proposed method took about 200 ms to generate an output image.

## 5. Conclusions

In this work, a new method to exploit the second-order statistics of encoded features in image style transfer is proposed. The proposed method matches the feature statistics of a content image into those of a target-style image under the assumption that the style features of a certain style have a distribution of multivariate correlated Gaussian. In the new feature transform layer, the second-order statistics (covariance and mean) of the encoded feature map of the content image are used to normalize the style of the content image, and the second-order statistics of the target style image are used to stylize the normalized feature to the target style image. Additionally, it is possible to control the style strength component-wise in the proposed feature transform layer, and, in doing this, various style combinations of images are achieved by using average styles and style components.

A new style loss, i.e., the sum of the covariance loss and mean loss, is proposed to minimize the possibility of style conflict that two different styles have the same Gram matrix. Incorporated with a fully trainable decoder network, an arbitrary style transfer function can be trained. The experimental results showed that considering correlation in a loss function improved the quality of the stylized image and further with correlation-aware feature transformation by eliminating the inconsistent patterns on the output image, which frequently appear in the previous methods with uncorrelated feature transform. By comparing the losses for a varying number of embedded styles, we demonstrated that using both the proposed feature transform method and style loss increased the style capacity of style transfer networks.

However, there are many interesting topics to deeply research regarding image style transfer, such as exploiting multi-scale responses, upgrading the feature transform layer to make end-to-end learning possible through the whole feed-forward network, matching the arbitrary distribution of features beyond second-order statistics and making a more general feature transform or decoder network to increase the style capacity of the networks.

## Figures and Tables

**Figure 1 sensors-22-02611-f001:**
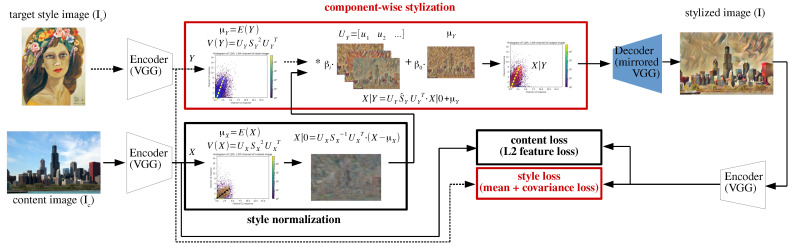
Architecture of the component-wise style transfer network: The solid lines and dashed lines represent the processing flow of the target style image and content image respectively. The two red boxes show the proposed modules, i.e., a component-wise stylization module and a new style loss.

**Figure 2 sensors-22-02611-f002:**
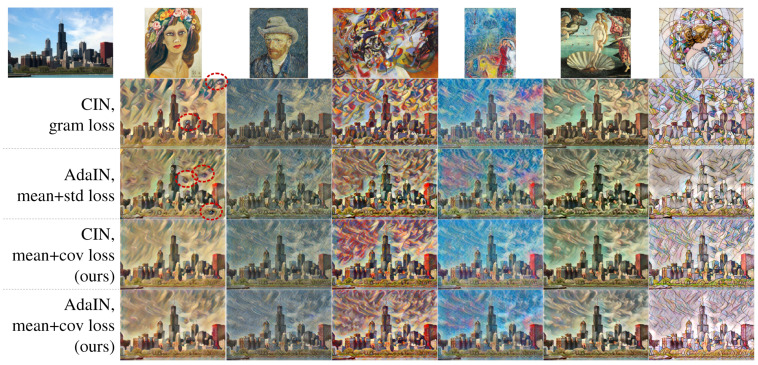
Style-transferred images in several seen styles with networks trained on a small style dataset. The red circles show artifacts on the output images.

**Figure 3 sensors-22-02611-f003:**
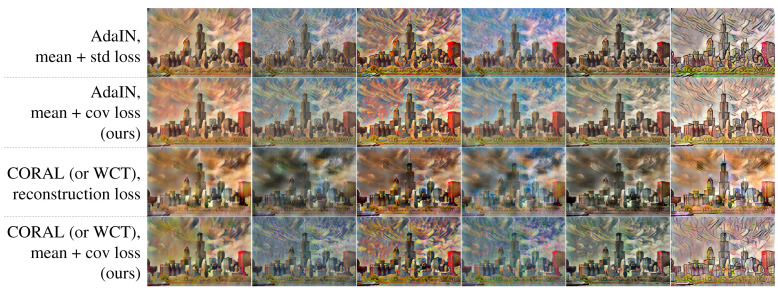
Style-transferred images into several unseen styles with networks trained on a large style dataset.

**Figure 4 sensors-22-02611-f004:**
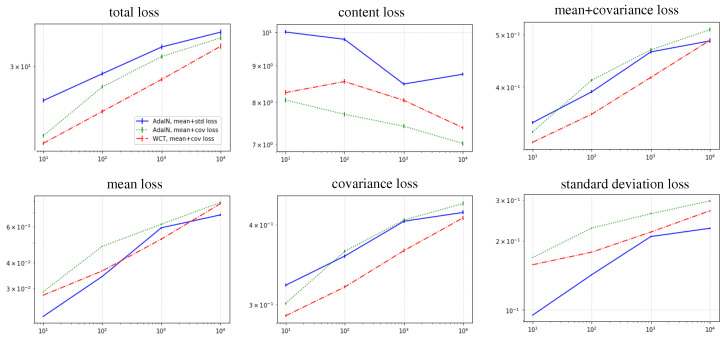
Comparison of the network capacity: loss performance versus the number of training styles.

**Figure 5 sensors-22-02611-f005:**
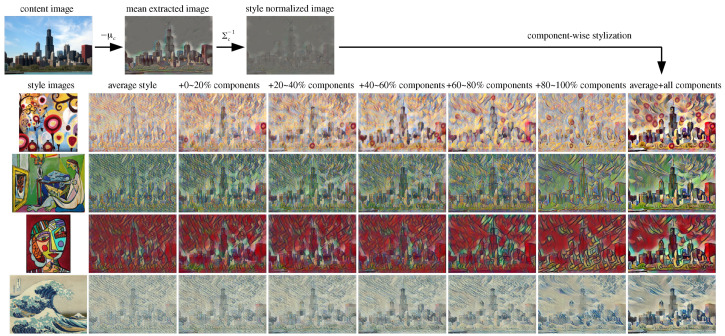
Examples of style-specific average and style components: The images on the first row represent the decoded images of features in style normalization procedure. The remaining rows of images show the average stylized images and their variations with some style components corresponding to each style image. The images on the third column are with style components corresponding to the lower 20% of the total eigenvalues of feature covariance and the images on the seventh column corresponding to higher 20%. The lower style components make subtle changes in the stylized image while the higher style components change the style of image largely from the average style. The right-most column of images represents the fully stylized images.

**Figure 6 sensors-22-02611-f006:**
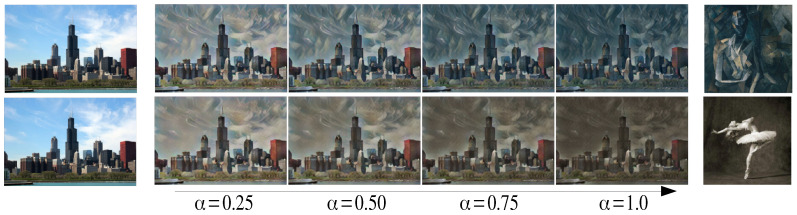
Examples of style strength control.

**Figure 7 sensors-22-02611-f007:**
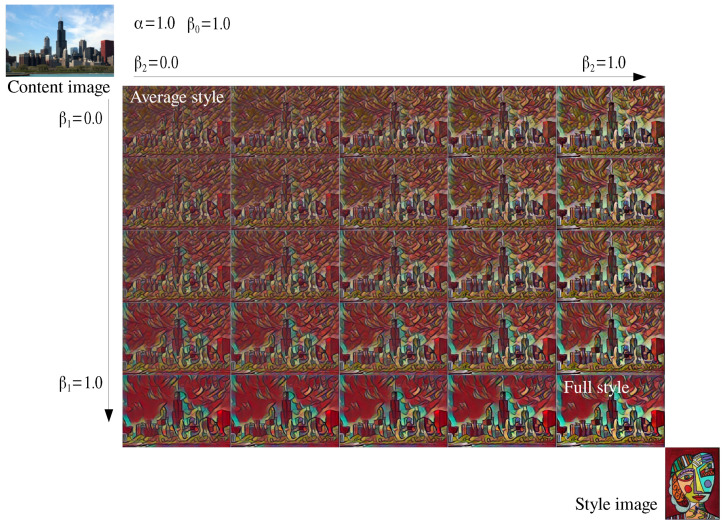
Examples of component-wise style control.

**Figure 8 sensors-22-02611-f008:**
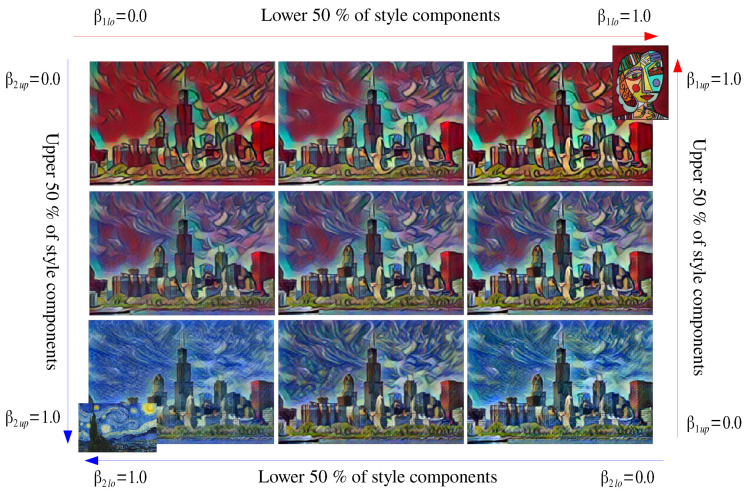
Examples of the component-wise style interpolation of two styles.

## Data Availability

Not applicable.

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
