# Peer review of "Toward Exploiting Second-Order Feature Statistics for Arbitrary Image Style Transfer"

_sensors, 2022, doi:10.3390/s22072611_

Round 1

Reviewer 1 Report

Review of article Sensors 1628986

The author proposes a new approach that realizes image style transferring using second-order statistics of the encoded features to build an image style transferring technique. He proposes to use a new correlation-aware loss as well as a correlation-aware feature alignment technique. As a result, the proposed method can generate various styles from one or several style images by using style-specific components from the second-order feature. The results of the new method are very convincing that the proposed loss function and alignment technique are working well. Nonetheless, I propose to put on the paper some execution times so that the asseveration of real-time processing capability can be demonstrated.

Please check the use of the article “the”. Also, there are several writing errors, please correct them. Here some of them: been improved in the quality of output; one or several input; achieved the multiple style; The two red boxes shows our; to generation; output image given input content; The first neural-network-based image; of neural style [2] is lack of style; a target style image; tranferring methods; control whole style; of target style images; The remained rows of images shows

Reviewer 2 Report

In the article, the author proposed a new method to exploit second-order statistics of encoded features in image style transfer. The first step includes a new correlation-aware loss as well as a correlation-aware feature alignment technique, followed by new component-wise style controlling method that can generate various styles from one or several input style images by using style-specific components.

Introduction is given in two paragraphs (Introduction and Background) Most of the references cited in this part are conference papers (17) dated at least 5 years ago. It would be desirable to update the literature. In the text(including abstract), in a couple of sentences it is written "we proposed, we show, ..." All of theese sentences should be rewritten in passive. 

Methods and experimental part are written clearly and are easy to follow. 

The images in the results are clear and legible. Only suggestion is to change the paragraphs 4.2 and 4.3 into paragraph 5 results, and therefore change the numbering of the conclusion to 6. 

I suggest to:

Accept after minor revision (corrections to minor methodological errors and text editing)

Reviewer 3 Report

This paper presents 'Toward Exploiting 2nd-order Feature Statistics for Arbitrary Image Style Transfer'. The idea is good. However, there are a few issues:

  1. Few of the abbreviations are not defined, for example, GRU etc. See abstract.
  2. One author wrote this paper, usage of 'we' in the paper does not make sense. Please avoid the first form. 
  3. Discuss numeric values in the abstract. 
  4. VGG is not defined.
  5. Few captions are very long. Please check this.
  6. Many sentences are vague, for example, 'However, their feature'. their etc should be avoided. Moreover, ' It used a pre-', why it etc is used on the paper. Please discuss the exact stuff.
  7. A lot of 'we' etc are used. Please avoid it. 
  8. Please double-check, a few parameters in equations are not defined.
  9. The same network is used (referred to in Ref 11). Please discuss the novelty.
  10. Again, 'component-wisely in our feature transform layer', our etc. must be avoided.

Round 2

Reviewer 3 Report

Paper is revised. I would recommend for publication.